



# The sea level time series of Trieste, Molo Sartorio, Italy (1869–2021)

Fabio Raicich[1]

[1]CNR, Institute of Marine Sciences, Trieste, I-34149, Italy

*Correspondence to*: Fabio Raicich (fabio.raicich@ts.ismar.cnr.it)

**Abstract.** The sea level observations carried out at Trieste, Molo Sartorio, from 1869 to 2021, have been revised and updated. Information on the tide gauges and on the geodetic benchmarks on Molo Sartorio during that period have been collected. Basic quality checks have been applied. The hourly data for the 1917–1938 period, digitised from the original charts, have allowed to build a time series of hourly sea level heights from 1905 to 2021. Gaps up to 24 hours have been filled by interpolation. The errors affecting the monthly and annual mean sea levels have been estimated. The availability of monthly and annual means

prior to 1904 allowed to build a mean sea level time series spanning 153 years, characterised by linear trends of observed sea level of 1.39±0.15 mm yr$^{-1}$ and of inverse-barometer-corrected sea level of 1.46±0.12 mm yr$^{-1}$. A significant acceleration of 0.008±0.004 mm yr$^{-2}$ was estimated from the inverse-barometer-corrected sea level time series. This data set represents the most up-to-date data set of sea level observations and ancillary information relative to the tide-gauge station of Trieste, Molo Sartorio.

The data are available through SEANOE (https://doi.org/10.17882/62758; Raicich, 2022).

## 1 Introduction

Long time series of environmental observations are fundamental in climate studies. Among them, it is widely recognized that historical sea level records play a key role in the assessment of long-term sea level rise rate and acceleration. Some records date back to the 18th century (e.g. Woodworth, 1999; Wöppelmann et al., 2006; Raicich, 2015) and automatic recordings started

in the first decades of the 19th century (Matthäus, 1972). Nevertheless, the number of homogeneous and sea level records longer than a century is quite limited, and, moreover, they are unevenly distributed geographically. Therefore, there is an increasing demand for the rescue of historical sea level observations.

The relevance of sea level data archaeology, that is the discovery, recovery, quality-control and publication of historical data for different purposes, is recognised by the Global Sea Level Observing System (GLOSS) (Bradshaw et al., 2015;

UNESCO/IOC, 2020a). Recent efforts include, among others, the recovery of long time series in Australia (Hunter et al., 2003), Croatia (Međugorac et al., 2022), France and its overseas territories (Wöppelmann et al., 2006; 2008; 2014; Testut et al., 2010; Guriou et al, 2013), Germany (Dangendorf et al., 2013), Italy (Battistin and Canestrelli, 2006; Bruni et al., 2019), New Zealand (Hannah, 2004), Portugal (Araújo et al., 2013), Spain (Marcos et al., 2011; 2013; 2021), the United Kingdom and its overseas territories (Woodworth, 1999; Woodworth et al., 2010), and the United States (Talke et al., 2014; 2018; 2020;

Ray and Talke, 2019).

In Trieste a tide gauge was installed in 1859 on Molo (Pier) Sartorio and, thanks to the efforts made by many people belonging to different organizations, sea level data spanning over 150 years are today available, making it one of the few ultra-centennial time series in the Mediterranean Sea (Zerbini et al., 2017). The geographical position of present tide-gauge station is (45.6472° N, 13.7596° E) (Becker et al., 2002) and it is shown in Fig. 1c. In the past monthly and annual means were summarized, for

example, by Polli (1938; 1970), AOP (1939), Ferraro (1972), and Raicich (2007); those data are regularly updated in the data bank of the Permanent Service for Mean Sea Level (PSMSL; https://psmsl.org/data/obtaining/stations/154.php). Mazelle (1909–1917) reported the hourly data for 1905–1911.

This work aimed at recovering and making available the sea level data obtained at Trieste, Molo Sartorio, as well as the information on the instruments used for the observations, and the levelling data of the benchmarks inside and outside the tide-





gauge hut. The existing data and metadata have been thoroughly revised and previously unpublished data have been added to the existing time series.

Section 2 describes the historical evolution of the station and the elements relevant to sea level observation. The observations are discussed in Sect. 3. In Sect. 4 the long-term time series obtained from the revised data set is described, and some basic properties are discussed. Data availability is summarized in Sect. 5. Conclusive remarks are presented in Sect. 6.

## 2 Historical background and local setting

### 2.1 Installations and instruments

The earliest known measurements of sea level height in Trieste were made by the physician Leonardo Vordoni from 1782 to 1794, who was interested in studying the connection of tides and the course of diseases. Although the data set is remarkably long for the period, it mainly has a historical value due to its low quality, as discussed in Raicich (2020). On 23 July 1840 Vincenzo Gallo, the director of the local meteorological observatory, began measuring the sea level height in order to compute the establishment of the port and to make tidal predictions (Gallo, 1840); observations for 21–22 March 1844 could only be recovered (Gallo, 1844).

The systematic observation of sea level started on 16 October 1859, when the first self-recording float tide gauge became operational (Table 1). It was provided with a stilling well opened in the floor of a room in the northwest corner of the Finance Guard building, at the end of Molo Sartorio (Schaub, 1860). Figure 2 displays the tide gauge, seen from the front and from above, and a vertical section of the stilling well, which was connected to the open sea by a siphon (Chiolich-Löwensberg, 1865; 1866). Around 1860 a vertical tube with a hydrometric scale was fixed to the northeast side of the pier, to carry out direct sea level measurements for calibration (MGI, 1897).

That setting remained unchanged until 29 November 1924 when the observations were interrupted because the building that hosted the tide gauge was completely restructured. Unfortunately, no temporary tide gauge was put in operation in the meantime, and the observations were resumed on 30 June 1926 in a new tide gauge hut, built on the same pier approximately 20 m to the east of the previous installation (Fig. 3). The station was provided with a stilling well which communicated with the sea through a 40-cm long pipe. Since then, the hydrometric tube could be accessed through the floor of the new tide gauge hut. It still exists although is it impossible to say if it is the original one, however it is no longer used as the calibration measurements are made in the stilling well.

In 1961 the hut was enlarged and a new stilling well was built (Picotti, 1960); this is the present installation. Figure 4 shows the tide gauge hut in 2001 and four tide gauges, three of which were operational at that time.

Table 1 summarizes the instruments used since 1869 and their main technical characteristics; all the tide gauges are float instruments. The instruments with chart speed greater than 10 mm h$^{-1}$ have a 24-h rotation drum, the others a 7-day drum. Unfortunately, little original documentation exists and the literature is often confusing, therefore, the table was compiled solving contradictions as far as possible. The date of the instrument change between 1859 and 1864 is unknown.

During its long history, the tide gauge has been managed by several organizations, namely the Central Maritime Government (*Central Seebehörde/Governo Centrale Marittimo*) from 1859 to 1919, the Royal Geophysical Institute (*Regio Istituto Geofisico*) from 1919 to 1941, the National Research Council of Italy (*Consiglio Nazionale delle Ricerche*) from 1941 to 1945, the Ministry of Agriculture and Forestry (*Ministero dell'Agricoltura e Foreste*) from 1945 to 1985, and, again, the National Research Council of Italy from 1985 onwards. These changes are reflected in the complex situations of tide-gauge zeros and benchmarks, that are described below.



## 2.2 The tide-gauge zeros

Until 7 December 1910 at 17h the tide-gauge zero corresponded to the top edge of the hydrometric scale, which was also known as 'the pier edge', 'the zero-line of the hydrometer' and simply 'Molo Sartorio'. It coincided with benchmark (*Höhenmarke*, HM) HM 39 of the Austrian Military-Geographic Institute (*k. u. k. Militär-geographisch Institut*, MGI). The

positive versus of the hydrometric scale was downwards (e.g. Sterneck, 1905). This zero was named '*Zero-point of Molo Sartorio*' for the first time by Anonymous (1910ca) and it has generally been known as '*Zero Molo Sartorio*' (ZMS) starting from Polli (1938).

From 7 December 1910 at 18h to 10 December 1910 at 08h a temporary zero was defined at 1.64 m below ZMS; the positive versus remained downwards (Osservatorio Marittimo, 1910). On 10 December 1910 at 19h a new zero was adopted at 2.16 m

below ZMS, corresponding to the lowest height recorded since an unknown date to 1910, namely on 15 January 1907 at 16h; it was known as '*Zero Hopfner*' (ZH) (Hopfner, 1913). Since then the positive versus has been upwards.

The zero was changed again probably in August 1919. It was set to 1.50 m below the mean sea level of 1911, also known as '*Hopfner Mean Sea Level*', namely 1.123 m above ZH. The zero introduced in 1919 was named '*Zero Istituto Talassografico*' (ZIT) for the first time by Ferraro (1972) and is the present zero. It corresponds to 2.537 m below ZMS.

Figure 5 summarizes the scheme of the tide-gauge zeros and the relationships with the tide-gauge contact point (CP) and benchmark (BM).

### 2.3 The benchmarks

#### 2.3.1 The Austrian period

The two earliest surveys were performed by the MGI in 1876 and 1884 (MGI, 1885; 1896). Both involved the vertical

benchmark in the tide gauge room, identified as HM 1. The 'absolute' height of HM 1, namely 3.352 m, was defined by adding 1.118 m, which was the mean sea level (MSL) relative to the tide-gauge zero, and 2.234 m, that is the height difference between HM 1 and HM 39 obtained in the levelling survey of 1876 (MGI, 1885; 1892). The MSL was probably relative to 1869. During a survey in 1884, HM 1 – HM 39 = 2.2341 m (MGI, 1896). The height of HM 1 was taken as the base of the levelling networks of the Austrian-Hungarian empire and of several countries that became independent after its dissolution.

Note that MGI (1897), belonging to the same collection as MGI (1896), reports HM 1 – HM 39 = 2.224 m, because the adopted MSL was 1.128 m, measured in 1875. A levelling in 1904 gave HM 1 – HM 39 = 2.2347 m (Sterneck, 1905), coherent with the value of 1876; Sterneck himself stated explicitly that 2.224 m was wrong.

#### 2.3.2 The Italian period

The first survey was performed in 1926, when the Italian Military Geographic Institute of Florence (*Istituto Geografico*

*Militare Italiano*, IGMI) levelled the new benchmarks installed after the new (present) tide gauge hut was built. On Molo Sartorio the levelling involved the vertical benchmark (*caposaldo verticale*, CsV) CsV 53, the horizontal benchmarks (*caposaldo orizzontale*, CsO) CsO 53, CsO 53A and CsO 54 and the tide-gauge CP (CP1926). The heights were referred to the Zero of the IGMI defined on the basis of the MSL of 1884–1903 at Genoa (IZ1894) (IGMI, 1926). The ZMS was the only surviving Austrian benchmark and its height was connected to the others in 1927 (Spinello, 1927).

The IGMI carried out another survey in 1956 involving CsO 39 and CsO 39'; the latter was former CsO 54, which had been renamed. A new Zero of the IGMI levelling network was adopted, based on the Genoa MSL of 1937–1946 (IZ1942) (Salvioni, 1957).

After the works on the tide gauge hut in 1961, a new tide-gauge CP was installed in 1965 (CP1965), known as '*Piastrina Mareografica*' (PM), which is currently used for the direct calibration measurements.





Other levelling surveys were carried out by IGMI in 1977, involving CsO 39a, CsO 39c, CsV 39, CP1965 (IGMI, 1977), and CsO 39' (Lama and Corsini, 2000), and in 1989, involving CsO 39a, CsO 39c, CsV 39 and CP1965 (Lama and Corsini, 2000). The most recent survey dates back to 2002. It was required because new benchmarks were installed before the existing benchmarks became unusable due to the restoration works of the building near the tide gauge hut. The levelling involved CsO

39a, CsO 39c, CsO 39''', which is the new name of CsO 39', and CP1965 (Zambon, G., CNR, Institute of Marine Sciences, Venice, personal communication, 2008).

### 2.3.3 Merging the benchmark heights

Overall, three national references frames have been used to refer the benchmark heights during the period of sea level observations, namely the Austrian Zero AZ1869 and the two Italian Zeros IZ1894 and IZ1942. In order to obtain a

homogeneous time series of height data, the relationships between those Zeros should be known. Unfortunately, it was not possible to find unambiguous information about them in the literature or in public archives. Therefore, they were estimated as explained in Appendix A.

Table 2a summarizes the original and normalised heights of the benchmarks on Molo Sartorio obtained during the national levelling surveys. The normalization was made using Eq. (A16) and (A17). Moreover, a composite time series of benchmark

heights relative to IZ1942 (Table 2b), was obtained by merging those of CsO 54/39'/39''' for 1926–2002, with those for 1876–1884 relative to the virtual benchmark VZMS, defined as:

$$VZMS = ZMS - 0.0109 \, m \qquad\qquad (1)$$

Equation 1 is based on the height difference between ZMS and CsO 54 measured by the Hydrographic Office of the Water Magistrate of Venice (*Ufficio Idrografico del Magistrato alle Acque*, UIMA) in 1926 (Spinello, 1927). as well as the composite

time series. Figure 6 displays the heights of the benchmarks involved in the composite time series in the respective reference systems.

The heights of 1977 are slightly higher than in 1956 and 2002, probably as an effect of the ground deformation induced by the earthquake of May 1976 in Friuli region (Talamo et al., 1978), having the epicentre at about 90 km from Trieste and $M_w = 6.4$ (Finetti et al., 1979). The heights of 1989 are anomalous, as they are about 7 cm higher than both in 1977 and 2002 for reasons

that could not be discovered; they were not taken into account.

The linear trend of the composite time series of tide-gauge BM heights is $0.07\pm0.06$ mm y$^{-1}$, significant at p = 0.02 and corresponding to a total height variation of +9±7 mm from 1876 to 2002. If the height of 1977 is not included in the analysis, the trend becomes $0.05\pm0.04$ mm y$^{-1}$, significant at p = 0.01 and corresponding to a total height variation of +6±5 mm in 126 years. In this work the errors correspond to 5 % significance. The result is consistent with the known relative stability of Trieste

compared to the other coastal areas of the north Adriatic (Carbognin and Taroni, 1996; Carbognin et al., 2004).

### 3 The observations

The data from 1859 to 1904 could only be found in the literature, except a few original charts covering about 10 days of August 1864 (CNA, 1864). The monthly and annual MSL for 1875–1904 are also summarised and discussed in Raicich (2007).

From 1905 onwards the data could be retrieved from original documents, namely tabulations of hourly heights for 1905–1911

and 1913–1914, and charts from 1917 onwards. Polli (1938) stated that manuscript hourly heights from 1912 to March 1915 did exist, and reported the monthly and annual means of 1912; the monthly and annual means of 1915 appeared for the first time in Polli (1970). Moreover, according to Osservatorio Marittimo (1916; 1917), the observations were regularly carried out in 1916 too, but they are missing.

Polli (1947) provided annual means for 1890–1904 but the data source was not quoted and it is unclear if they came from

observations or were estimated. Moreover, the values of 1901–1904 are different from those in Sterneck (1905) who explicitly



reported that he obtained the data from the Maritime Observatory (*Osservatorio Marittimo*). Therefore, we considered Polli's data suspect and discarded them.

## 3.1 The data since 1905

The complete time series of hourly data from 1905 to 2021 obtained in this work is available in Raicich (2022). The data for 1905–1914 have been digitised from the original tabulations. Due to the absence of original charts for a direct check, only evident mistakes could be corrected. The hourly heights from 1917 to 1938 have been digitised from the original charts; previously, only high waters (HWs) and low waters (LWs) were available. The hourly data from 1939 onwards, that were already available, have also been thoroughly revised.

In principle, the hourly heights are 'instantaneous', that is the curves were not filtered before digitisation. However, in case of oscillations of period shorter than approximately 10–15 minutes, the persons in charge of data digitisation used to smooth the curve graphically and digitise the smoothed values.

Gaps no longer than 24 hours were filled by interpolation according to UNESCO/IOC (1994; 2020b), that is using linear interpolation of the residual sea level, obtained after subtraction of the astronomical tide from the observations. In case of failure of the main tide gauge, the sea level heights were generally taken from the charts of auxiliary instruments characterised by 7-day rotation drum, such as the R 225 and the Richard (Table 1). This allowed to digitise the HWs and LWs but made it difficult to extract the hourly values which, in fact, were not usually reported. In such cases, we obtained the hourly values by means of cubic spline interpolation of the HW and LW data. In order to treat the HWs and LWs as true local extremes, two auxiliary data were introduced, one minute before and one minute after each extreme, respectively; they are 1 mm lower/higher than the corresponding HW/LW. This procedure ensures that the estimated values do not overshoot/undershoot the observed local extremes. We stress that interpolation aims at obtaining reasonable hourly values for the estimate of daily MSL and the subsequent calculation of monthly and annual means, not at estimating the missing data.

The only major gaps are in 1912 and 1915–1916, due to missing observations as explained above, and from 29 November 1924 to 30 June 1926, when the tide gauge was dismantled and reinstalled in the new hut. The other gaps that occurred for different reasons and that could not be filled are summarized in Appendix B.

From 1905 to 2021, except the December 1924 – June 1926 period in which the tide gauge was not operational, 1011736 hourly values are potentially available. The number of those estimated by interpolation is 3531, corresponding to 0.3 %, while 30883, that is 3.1 %, are missing. At least one auxiliary tide gauge became available in 1927 (Table 1), and this allowed to reduce the missing hourly data to 0.1 % since then.

From the hourly data, daily MSL were estimated by means of Doodson X0 filter. Monthly MSLs were computed when at least 50 % of the daily values were available. As a result, the monthly MSL could not be determined for the following months: January – December 1912, January 1915 – December 1916, and December 1924 – June 1926. Annual (calendar) MSLs were computed with at least 11 monthly means, therefore, they could not be determined for 1912, 1915–1916, 1925, and 1926. For 1912 and 1915, we adopted the monthly and annual MSLs available from the literature (Polli, 1938; 1970).

## 3.2 The assessment of errors

It is not easy to associate errors to the hourly heights digitised from charts, therefore, we only aim at estimating representative values.

Because the sea level height is defined on the basis of the vertical distance between the tidal curve and a baseline drawn on the chart, we must take into account the accuracy of the positions of those lines. The baseline of the Fuess-Seibt tide gauge charts was identified by a horizontal line drawn *a posteriori* 8 mm above the bottom edge of the chart. The Ott-Büsum tide gauge provided both the tidal curve and the baseline. At least before June 1961 the curve of the Fuess-Seibt was originally marked by a metal tip on special coated paper. The curve was very thin and often faint, therefore, the persons in charge of data

digitisation used to trace it with a coloured pencil or ink; that practice might have introduced errors but it made it possible to distinguish each day in case of overlapping curves, as the paper was generally changed every two or three days. In any case, the line thickness of each line is about 0.5 mm. As the reduction ratio is 1/10, it is realistic to associate a 1-cm accuracy to the individual digitised heights. We also recall that, because the charts before 1917 are not available, we could not verify the

accuracy of the 1905–1914 data.

The uncertainty associated to the interpolation of gaps using cubic splines was assessed empirically as follows. As HW and LW data are available for 1917-2021, we estimated a time series of hourly heights for that period using cubic splines (as in Sect. 3.1). The root-mean-square difference between estimated and observed hourly values is about 5 cm, which was assumed as the representative uncertainty on the individual hourly height estimated with splines. The uncertainty related to the

interpolation based on de-tided residuals is more difficult to assess, because the procedure is intrinsically more complex, as it involves the estimate of the astronomical tide, the subtraction from the observations and the interpolation on a specific time interval. Because we only aim at representative errors, in this case we also assumed a 5-cm uncertainty on the individual estimated hourly height. The errors on the daily, monthly and annual MSL were estimated on the basis of the actual number of interpolated hourly data involved.

As a result, the daily MSL is affected by an error between 0.15 cm (no interpolated hourly data) and 0.77 cm (24 interpolated hourly data). Figure 7 displays the monthly errors and percentages of valid days until 1975; afterwards, there is no missing data. With regard to the 1905-2021 period, the MSL of a month with no missing days is affected by an error of 0.03 cm; the largest monthly error is 0.09 cm, due to missing days and/or daily means estimated from interpolated hourly values (Fig. 7a). The errors on annual MSLs are always 0.01 cm (to the centimetre precision). It is reasonable to adopt these errors also for the

monthly and annual MSLs of 1901–1904, 1912 and 1915, which for which the amounts of valid days are unknown.

The original monthly means of the 1875–1889 period are Mean Tide Levels (MTLs). MSLs were estimated using average monthly corrections obtained by comparing the MTLs and MSLs of 1917–2021 (Table 3), thanks to the availability of HWs and LWs during that period. The typical error on monthly MTL is around 0.10 cm (between 0.09 and 0.11 cm), depending on data availability (Fig. 7b). The errors on the estimated monthly MSLs range approximately between 0.4 and 0.6 cm and are

mostly determined by the error on the corrections (Fig. 7a). The errors on the annual MSLs for 1875–1884 are 0.14 cm, including the corrections from MTL to MSL (Table 3); this value is also representative of the error for 1885–1889.

## 4 The long-term Mean Sea Level time series

Figure 8 displays the time series of monthly MSL (a) and annual MSL (b), relative to ZIT.

The time series reflects the behaviour of sea level variability common to the Mediterranean Sea stations, coherent with the

global sea level rise, except for a period of stability approximately between the 1960's and the early 1990's (Tsimplis and Baker, 2000; Marcos and Tsimplis, 2008; Gomis et al., 2012; Zerbini et al., 2017).

The digitisation of the 1917–1938 hourly data and the revision of the whole time series mostly led to minor differences from the previously known monthly means, but there are some exceptions. Appendix C summarizes the main differences between the values obtained in this work and those in the PSMSL data bank, used for reference.

There is an issue on the annual mean of 1869. The MSL reported in Lorenz et al. (1873) corresponds to 1.407 m above ZIT, and this has been verified to be correct using the data therein. On the other hand, the MSL of 1869, used to define the Zero of the Austrian levelling network, is 1.118 m below ZMS (MGI, 1885; 1892; see Sect. 2.3.1), that is 1.419 m above ZIT. The reason for the difference is unknown, but it could be in the conversion from the Viennese foot (the original unit) to the SI units and subsequent rounding or truncations.

In practice, the MSL in Lorenz et al. (1873) is homogenous with the rest of the sea level data that are referred to ZIT, while the MSL in MGI (1885; 1892) represents the height of ZMS in the Austrian levelling reference system and is involved in the



time series of the benchmarks heights shown in Fig. 5. A problem might arise only if the two reference systems interacted with each other, which is not the case in this work.

The linear trend computed with all the annual MSLs from 1869 to 2021 (Fig. 8b) is $1.36\pm0.17$ mm yr$^{-1}$. Taking into account only the period covered by hourly data (1905–2021), the trend is the same, namely $1.36\pm0.19$ mm yr$^{-1}$. Note that in the fit the

annual MSLs have been weighted with the inverse of the respective errors (see Sect. 3.2).

Among the factors that affect the sea level variability there is the inverted-barometer (IB), which consists in an inverse relationship between variations of atmospheric pressure and sea level. In equilibrium conditions, 1 hPa of atmospheric pressure increase approximately corresponds to 1 cm of sea level decrease, and *vice versa*. According to Raicich and Colucci (2021a) the atmospheric pressure at Trieste exhibits a significant linear trend during the last 150 years, namely $0.5\pm0.2$ hPa per century.

The standard IB correction ($-1$ cm hPa$^{-1}$) was applied to the MSL time series using the pressure data in Raicich and Colucci (2021b). As a result, for the IB-corrected MSL we obtain a linear trend of $1.45\pm0.13$ mm yr$^{-1}$ for 1869–2021, and $1.46\pm0.15$ mm yr$^{-1}$ for 1905–2021.

The linear trends estimated here are slightly lower than the global value of $1.73\pm0.44$ mm yr$^{-1}$ in Fox-Kemper et al. (2021) for 1901–2018, but within the interval defined by the uncertainties.

The linear trends for 1993-2021, that is the period covered by satellite altimetry, are $3.03\pm0.14$ mm yr$^{-1}$ (observed sea level), and $3.02\pm0.11$ mm yr$^{-1}$ (IB-corrected sea level), which turn out to be slightly lower than the global value of $3.4\pm0.4$ mm yr$^{-1}$ (https://sealevel.nasa.gov; Beckley et al., 2017; 2021), but consistent within the uncertainties.

The sea level acceleration was estimated as twice the coefficient of the quadratic term of the second-order polynomial fit. If the observed sea level is taken into account, for 1869–2021 the acceleration is $0.006\pm0.005$ mm yr$^{-2}$ (p = 0.23 significance)

and for 1905–2021 it is $0.008\pm0.006$ mm yr$^{-2}$ (p = 0.22). From the IB-corrected sea level, the acceleration is $0.008\pm0.004$ mm yr$^{-2}$ (p = 0.05 significance) for 1869–2021, and $0.009\pm0.005$ mm yr$^{-2}$ (p = 0.06) for 1905–2021. These values are included  in the range of ($-0.002$, $+0.019$) mm yr$^{-2}$, relative to 1902–2010, adopted in Fox-Kemper et al. (2021).

## 5 Data availability

The hourly sea level data and the derived monthly and annual mean sea levels used in this work are available from SEANOE

as "Raicich, F.: Sea-level observations at Trieste, Molo Sartorio, Italy", (https://doi.org/10.17882/62758; Raicich, 2022).

## 6 Summary and conclusions

We have revised and updated the information about the sea level observations carried out at Trieste, Molo Sartorio, from 1869 to 2021, using the material in the archive of CNR, Institute of Marine Sciences (ISMAR), Trieste, and published and unpublished documents from other sources.

We could identify the tide gauges used for the observations, and we could recover the heights of the geodetic benchmarks on Molo Sartorio since the late 19[th] century.

The digitisation of hourly data from 1905 to 1938 from the original tabulations or charts allowed to extend the available time series from 1905 to 2021. The quality control provided information on data gaps, most of which were filled by interpolation in order to obtain reasonable daily sea level means and, consequently, compute reliable monthly and annual MSLs. The

monthly and annual MSLs obtained from hourly data are affected by errors of 0.01 and 0.03 cm, respectively.

Monthly and annual MTL from 1869 to 1889 and MSLs for 1901–1904 were also available from the literature, although with gaps. Thanks to average monthly corrections (Table 3) the MTLs were normalised to MSLs, which allowed to build monthly and annual MSL time series spanning 153 years. The linear trend of the 1869–2021 observed sea level is $1.39\pm0.15$ mm yr$^{-1}$, that of the IB-corrected sea level is $1.46\pm0.12$ mm yr$^{-1}$. The acceleration estimated from observed sea level is $0.006\pm0.005$ mm





yr$^{-2}$, and from IB-corrected data is 0.008±0.004 mm yr$^{-2}$. Both the linear trends and the acceleration are consistent with long-term global estimates Fox-Kemper et al. (2021).

The outcome of this sea level data archaeology work represents the most up-to-date data set of sea level observations and ancillary information relative to the tide-gauge station of Trieste, Molo Sartorio.

## 5   Appendix A: The relationships between the Austrian and Italian levelling Zeros

Let us recall the definitions: AZ1869 is the Austrian Zero based on Trieste MSL of 1869, IZ1894 is the Italian Zero based on Genoa MSL of 1884–1903, and IZ1942 is the Italian Zero based on Genoa MSL of 1937–1946. The available relationships are:

$$ZMS - AZ1869 = 1.110 \; m \tag{A1.1}$$

$$ZMS - AZ1869 = 1.118 \; m \tag{A1.2}$$

$$HMSL - ZMS = -1.037 \; m \tag{A2}$$

$$CsO\,54 - IZ1894 = 1.1112 \; m \tag{A3}$$

$$CsO\,54 - ZMS = -0.0109 \; m \tag{A4.1}$$

$$CsO\,54 - ZMS = -0.0113 \; m \tag{A4.2}$$

$$CM1825 - HMSL = 0.159 \; m \tag{A5}$$

$$ZMPS - CM1825 = -0.2246 \; m \tag{A6}$$

$$CsO\,39' - IZ1942 = 0.8567 \; m \tag{A7}$$

$$IZ1942 - ZMPS = 0.2356 \; m \tag{A8}$$

Equations (A1.1) and (A4.2) were taken from Morelli (1950). Eq. (A1.2) from MGI (1892; 1896), Eq. (A2) from Hopfner (1913), Eq. (A3) from IGMI (1926), Eq. (A4.1) from Spinello (1927), Eq. (A5) from UIMA (1941), Eq. (A6) from Dorigo (1961), Eq. (A7) from IGMI (1967), and Eq. (A8) from Cavazzoni (1977).

ZMS is 'Zero Molo Sartorio' and HMSL is 'Hopfner MSL' (Sect. 2.2). CsO 54 and CsO 39' represent the same benchmark, which was renamed (Sect. 2.3.2). CM1825 is the 'Comune Marino' (CM) of 1825 at Venice; it is identified by the upper edge of the algae belt that forms on quays, and approximately corresponds to the wetting caused by the combination of high tides and waves (Camuffo, 2017). The CM has represented the local levelling reference plane until the late 19th century. ZMPS is 'Zero Mareografico Punta Salute', that is the MSL at Venice of 1884–1909, central year 1897, used as the tide-gauge datum since 1923.

A major ambiguity concerns Eqs. (A1.1) and (A1.2). The values represent the annual MSL measured downwards relative to ZMS, which was used to define the Austrian Zero. Unfortunately, the origin of the MSL values of 1.110 and 1.118 below ZMS is partly unclear. Equation (A1.1) was adopted by Morelli (1950) who quoted Lorenz et al. (1873) as the data source. Using also Eq. (A3) and (A4.1), Morelli estimated:

$$AZ1869 - IZ1894 = 0.012 \; m \tag{A9}$$

Actually, the data in Lorenz et al. (1873) lead to a MSL of 1.130 m below ZMS. The difference could be explained by the conversion from the Viennese foot (the original unit) to the SI units and subsequent rounding or truncations. On the other hand, MGI (1885; 1892) report values of 1.12 and 1.118 m, respectively, and MGI (1896) reports 1.1179 m, as a result of subtracting 2.2341 m from 3.3520 m (Sect. 2.3.1). Here, Eq. (A1.2) was preferred because the MGI publications were considered official. A lesser ambiguity concerns Eqs. (A4.1) and (A4.2). Eq. (A4.1) was adopted here because the measurement was carried out shortly after the IGMI levelling of 1926, and because in 1949 it was rather difficult to access ZMS (Morelli, 1950). The difference between AZ1869 and IZ1894 was computed from Eqs. (A1.2), (A3) and (A4.1):

$$AZ1869 - IZ1894 = 0.0041 \; m \tag{A10}$$




Recalling that CsO 54 was renamed CsO 39', the difference between IZ1942 and IZ1894 was computed from Eqs. (A3) and (A7):

$$IZ1942 - IZ1894 = 0.2545\ m \tag{A11}$$

As a consequence, from Eqs. (A10) and (A11):

$$IZ1942 - AZ1869 = 0.2504\ m \tag{A12}$$

Another approach is possible. By adding together Eqs. (A1.2), (A2), (A5), (A6) and (A8):

$$IZ1942 - AZ1869 = 0.2510\ m \tag{A13}$$

and, using Eq. (A10):

$$IZ1942 - IZ1894 = 0.2551\ m \tag{A14}$$

There is a 0.0006-m difference between Eq. (A13) and Eq. (A12), and between Eq. (A14) and Eq. (A11), but a 0.1-mm precision is unrealistic because only Eqs. (A3), (A4.1) and (A7) are results of levelling, while a MSL is involved in the other equations. Therefore, we adopted:

$$AZ1869 - IZ1894 = 0.004\ m \tag{A15}$$

$$IZ1942 - IZ1869 = 0.251\ m \tag{A16}$$

$$IZ1942 - IZ1894 = 0.255\ m \tag{A17}$$

where the value in Eq. (A16) is the average of those in Eqs. (A12) and (A13), and the value in Eq. (A17) is the average of those in Eqs. (A11) and (A14).

**Appendix B: The gaps in the hourly time series from 1905 to 2021.**

As explained in Sect. 3.1, the hourly data gaps up to 24 hours were filled by interpolated to obtain reasonable daily mean sea levels. The gaps that could not be filled are summarized in Table B1, alongside the reasons that caused the observations to be missed. They were grouped into four categories: 1) Mechanical malfunctions; 2) station maintenance or repairs; 3) Recording or chart problems; 4) other known reasons; 5) unknown reasons.

**Table B1. List of the gaps in the hourly time series.**

| Start and end dates | Missing hours | Reason |
|---|---|---|
| **1) Mechanical malfunctions** | | |
| 24/09/1906–03/10/1906 | 232 | clock repair |
| 25/12/1907–03/01/1908 | 212 | clock repair |
| 23/01/1911–26/01/1911 | 69 | float problems |
| 04/02/1911–06/02/1911 | 43 | clock problems |
| 05/01/1918–08/01/1918 | 72 | clock stop |
| 19/11/1918–20/11/1918 | 32 | clock stop |
| 13/11/1919–14/11/1919 | 29 | clock stop |
| 01/06/1920–02/06/1920 | 30 | loose screw |
| 26/07/1920–29/07/1920 | 79 | loose screw |
| 02/09/1920–04/09/1920 | 47 | clock stop |
| 20/08/1921–22/08/1921 | 54 | tide-gauge failure |
| 28/10/1922–30/10/1922 | 48 | clock failure |
| 30/06/1923–03/07/1923 | 72 | clock failure |
| 30/10/1926–03/11/1926 | 96 | tide gauge failure |
| 03/12/1926–09/12/1926 | 156 | tide gauge failure |
| 10/11/1927–14/11/1927 | 96 | tide gauge failure |
| 12/03/1928–16/03/1928 | 93 | tide gauge failure |
| 17/04/1929–20/04/1929 | 71 | clock failure |
| 01/09/1934–03/09/1934 | 41 | clock stop |
| | | |
| **2) Station maintenance or repairs** | | |
| 28/09/1908–11/10/1908 | 312 | tide gauge maintenance |
| 10/01/1913–13/01/1913 | 69 | tide gauge repair |
| 15/04/1961–17/04/1961 | 41 | tide gauge hut enlargement |
| 20/06/1966–23/06/1966 | 75 | stilling well maintenance |



3) Recording or chart problems

| | | |
|---|---|---|
| 28/12/1917–05/01/1918 | 192 | missing charts |
| 14/04/1918–21/04/1918 | 171 | unreadable curve |
| 30/12/1918–03/01/1919 | 98 | missing chart |
| 09/04/1919–13/04/1919 | 98 | missing chart |
| 09/08/1920–13/08/1920 | 97 | missing chart |
| 29/12/1922–31/12/1922 | 50 | missing chart |
| 24/08/1929–26/08/1929 | 48 | missing curve |
| 21/10/1933–23/10/1933 | 48 | missing chart |
| 30/09/1934–03/10/1934 | 61 | missing chart |
| 11/10/1960–12/10/1960 | 38 | missing tidal curve |

4) Other known reasons

| | | |
|---|---|---|
| 15/10/1908–18/10/1908 | 72 | chart blown away by the wind |
| 01/01/1912–01/01/1913 | 8784 | missing data tables |
| 01/01/1915–11/01/1917 | 17797 | missing data tables or charts |
| 29/11/1924–30/06/1926 | 13881 | tide gauge non operational |
| 30/04/1945–07/05/1945 | 162 | related to war events |
| 08/12/1954–11/12/1954 | 78 | unreliable baseline |

5) Unknown reasons

| | |
|---|---|
| 27/02/1905–01/03/1905 | 48 |
| 25/08/1906–28/08/1906 | 60 |
| 04/08/1907–12/08/1907 | 195 |
| 21/08/1907–23/08/1907 | 47 |
| 28/10/1907–31/10/1907 | 71 |
| 20/12/1907–21/12/1907 | 25 |
| 22/12/1907–25/12/1907 | 68 |
| 05/06/1909–07/06/1909 | 55 |
| 24/11/1909–26/11/1909 | 43 |
| 01/12/1909–02/12/1909 | 32 |
| 03/12/1909–05/12/1909 | 49 |
| 08/02/1910–10/02/1910 | 45 |
| 18/02/1910–20/02/1910 | 48 |
| 03/03/1910–04/03/1910 | 25 |
| 23/06/1911–25/06/1911 | 38 |
| 26/06/1911–29/06/1911 | 74 |
| 24/05/1965–28/05/1965 | 96 |

**Appendix C: Differences from the monthly means of this work and those in the PSMSL data bank**

Here we discuss how the updated monthly and annual time series differ from those available from the PSMSL data bank (https://www.psmsl.org/data/obtaining/stations/154.php). The RLR datum of PSMSL is 9.4 m below the tide-gauge CP, corresponding to 5.407 m below ZIT.

5 The difference between the monthly MSLs of this work and those of the PSMSL data bank are shown in Fig. C1, where departures greater than 1 cm are also highlighted.

A difference common to all the monthly MSLs is related to the method of calculation. Here, first daily MSLs have been estimated by applying Doodson X0 filter to the hourly values, then monthly values have been computed by averaging the daily MSLs. In the past the daily MSL was generally computed by averaging the hourly sea levels (the heights at 00h and 24h were given a weight of 0.5 and those from 01h to 23h a weight of 1).

10 The main differences are:

1) The annual MSL for 1869 is reported for the first time.

2) The monthly MSLs for 1875–1889 are estimated using average monthly corrections (Table 3). By contrast, the data of PSMSL were estimated by subtracting 2 mm from the MTLs, regardless of the month. The original MTLs are here

15 confirmed, with marginal differences due to rounding, except those of October 1881, that was found to be wrong in the original source (Governo Marittimo, 1877–1890), and March 1884, that was previously reported erroneously (Raicich, 2007).

3) The MSLs for 1905–1911 and 1913–1914 were revised. Besides the method of computing the MSL, differences come



from the interpolation of gaps.

4) Monthly and annual MSLs are now available for 1917–1938, while, previously, they were only available for 1920–1922. The PSMSL data bank does not include the data for 1917, 1918, 1923 and July–December 1926; instead, it includes December 1924, which does not exist.

5) The monthly MSLs of November–December 1922 and January–November 1924 were substantially corrected, after revising the relationship between the zero-level of the charts and the tide-gauge zero.

6) The same revision was done for December 1954–June 1955, that, according to the PSMSL, were anomalous compared to nearby stations. The corrected data were checked to be consistent with those of Venice Punta Salute (data from Battistin and Canestrelli, 2006), Falconera (UIMA, 1924–1925) and Porto Lignano (UIMA, 1956) (Fig. 1).

**Author contribution**

FR built the time series, carried out the data quality control, and wrote the paper.

**Competing interests**

The author declares that he has no conflict of interests.

**Acknowledgements**

The author thanks the Italian Military Geographic Institute of Florence for giving access to historical levelling data of benchmarks on Molo Sartorio.

The author acknowledges the work done by the previous staff of the Maritime Observatory, the Geophysical Institute, the Thalassographic Institute, and the Institute of Marine Sciences, who managed and maintained the tide gauge, carried out the observations, processed and preserved the data used to build the sea level data set. The author would like to thank M. Iorio and E. Caterini, in the current staff of the Institute of Marine Sciences of CNR, and R.R. Colucci, previously at the Institute of Marine Sciences and now at the Institute of Polar Sciences of CNR.

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



**Table 1. Instruments used in the tide-gauge station and sources of their technical characteristics. R is the reduction ratio; V is the paper speed. A dash (–) indicates that the model is unknown. The information in square brackets is uncertain.**

| Period | Manufacturer / model | Analogue | | Digital |
|---|---|---|---|---|
| | | R | V (mm h$^{-1}$) | Sampling rate |
| **Main instruments** | | | | |
| 16/10/1859–20/12/1859 | Müller und Schild (Trieste, Austria) / – | 1/6.43 | 28 | |
| 11/08/1864–07/12/1910 | Stabilimento Tecnico Triestino (Trieste, Austria) / – | 1/6.43 | 35 | |
| 10/12/1910–08/06/1961 | R. Fuess (Berlin, Germany) / Seibt No. 605 | 1/10 | 30 | |
| 08/06/1961–31/12/1961 | Thalassia (Trieste, Italy) / – | 1/10 | 12 | |
| 31/12/1961–29/10/1984 | R. Fuess (Berlin, Germany) / Seibt No. 605 | 1/10 | 30 | |
| 04/10/1966–present | A. Ott (Kempten, Germany) / Büsum | 1/10 | 20 | |
| 11/01/2001–30/05/2017 | A. Ott (Kempten, Germany) / Thalimedes (1st) | | | 1 min$^{-1}$ |
| 30/05/2017–16/04/2021 | A. Ott (Kempten, Germany) / Thalimedes (3rd) | | | 1 min$^{-1}$ |
| 16/04/2021–present | A. Ott (Kempten, Germany) / Thalimedes (4th) | | | 1 min$^{-1}$ |
| **Auxiliary Instruments** | | | | |
| 1927–? | Officina Meccanica di Precisione (Stra, Italy) / R 225 | [1/5] | [2] | |
| 1927–??/08/1966 | Officina Meccanica di Precisione (Stra, Italy) / M 450 | 1/5 | 15 | |
| [1928]–[1953] | Richard (Paris, France) / – | 1/10 | 2 | |
| 08/10/1969–05/09/1970 | Thalassia (Trieste, Italy) / – | 1/10 | 4.5 | |
| 31/01/1972–04/12/2013 | Thalassia (Trieste, Italy) / – | 1/10 | 2 | |
| 14/01/2005–16/04/2021 | A. Ott (Kempten, Germany) / Thalimedes (2nd) | | | 1 min$^{-1}$ |
| 16/04/2021–present | A. Ott (Kempten, Germany) / Thalimedes (3rd) | | | 1 min$^{-1}$ |

**Table 2. a) Heights (m) of the benchmarks near the tide gauge at Molo Sartorio, relative to IZ1942. The measurements in the original reference system, namely AZ1869 for 1876 and 1884, and IZ1894 for 1926, are reported in brackets. b) Composite time series of heights (m) of VZMS = ZMS – 0.0109 m, and CsO 54/39'/39''', relative to IZ1942. TG BM is the tide-gauge benchmark, TG CP is the tide-gauge contact point. See the text for other details on denominations and normalizations.**

| | 1876 | 1884 | 1926 | 1956 | 1977 | 1989 | 2002 |
|---|---|---|---|---|---|---|---|
| **a)** | | | | | | | |
| HM 1 | 3.101 (3.352) | 3.1010 (3.3520) | | | | | |
| HM 39 = ZMS | 0.867 (1.118) | 0.8669 (1.1179) | | | | | |
| CsO 53 | | | 1.1627 (1.4177) | | | | |
| CsO 53A | | | 0.8983 (1.1533) | | | | |
| CsV 53 = CP1926 | | | 2.3670 (2.6220) | | | | |
| CsO 54/39'/39''' (TG BM) | | | 0.8562 (1.1112) | 0.8567 | 0.8669 | | 0.8629 |
| CsO 39a | | | | 0.8780 | 0.8861 | 0.9561 | 0.8799 |
| CsO 39b | | | | 0.5184 | | | |
| CsO 39c | | | | 0.6642 | 0.6723 | 0.7429 | 0.6666 |
| CsV 39 | | | | | 3.1997 | 3.2715 | |
| PM = CP1965 (TG CP) | | | | | 2.3332 | 2.4044 | 2.3310 |
| **b)** | | | | | | | |
| VZMS + CsO 54/39'/39''' | 0.8561 | 0.8560 | 0.8562 | 0.8567 | 0.8669 | | 0.8629 |

**Table 3. Average monthly and annual differences (centimetres) between MTL and MSL from 1917–2021 data.**

| | MTL–MSL |
|---|---|
| January | 0.30±0.61 |
| February | –0.02±0.50 |
| March | –0.17±0.54 |
| April | 0.03±0.53 |
| May | 0.36±0.40 |
| June | 0.40±0.43 |
| July | 0.21±0.44 |
| August | –0.06±0.43 |
| September | –0.27±0.40 |
| October | –0.14±0.45 |
| November | 0.16±0.56 |
| December | 0.40±0.54 |
| Year | 0.10±0.03 |

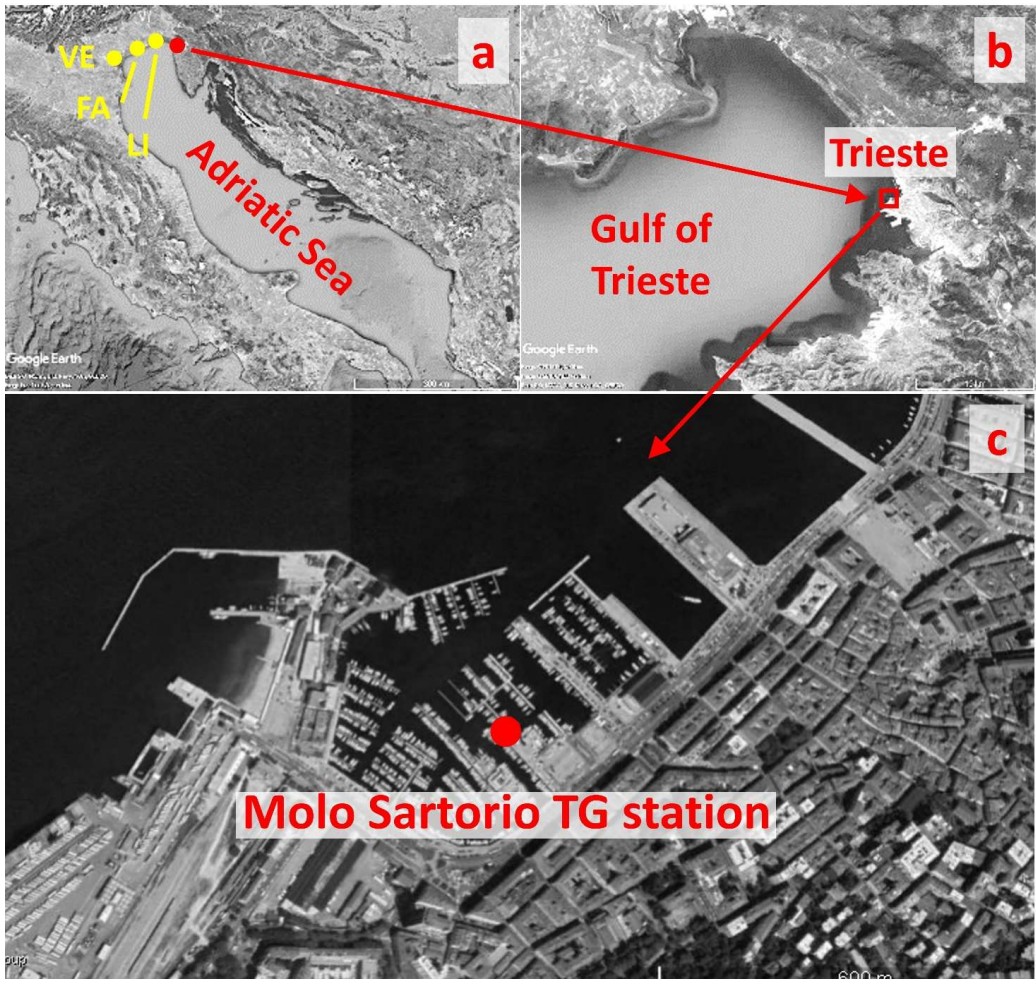

**Figure 1: (a) The Adriatic region. (b) The Gulf of Trieste. (c) Aerial image of Trieste harbour with the location of the tide-gauge (TG) station. The yellow dots in (a) indicate the stations of: Venice Punta Salute (VE), Falconera (FA), Porto Lignano (LI). (Images extracted from © Google Earth; © 2020 Landsat/Copernicus, © 2020 CNES/Airbus, © 2020 Digital Globe, © 2020 TerraMetric.)**





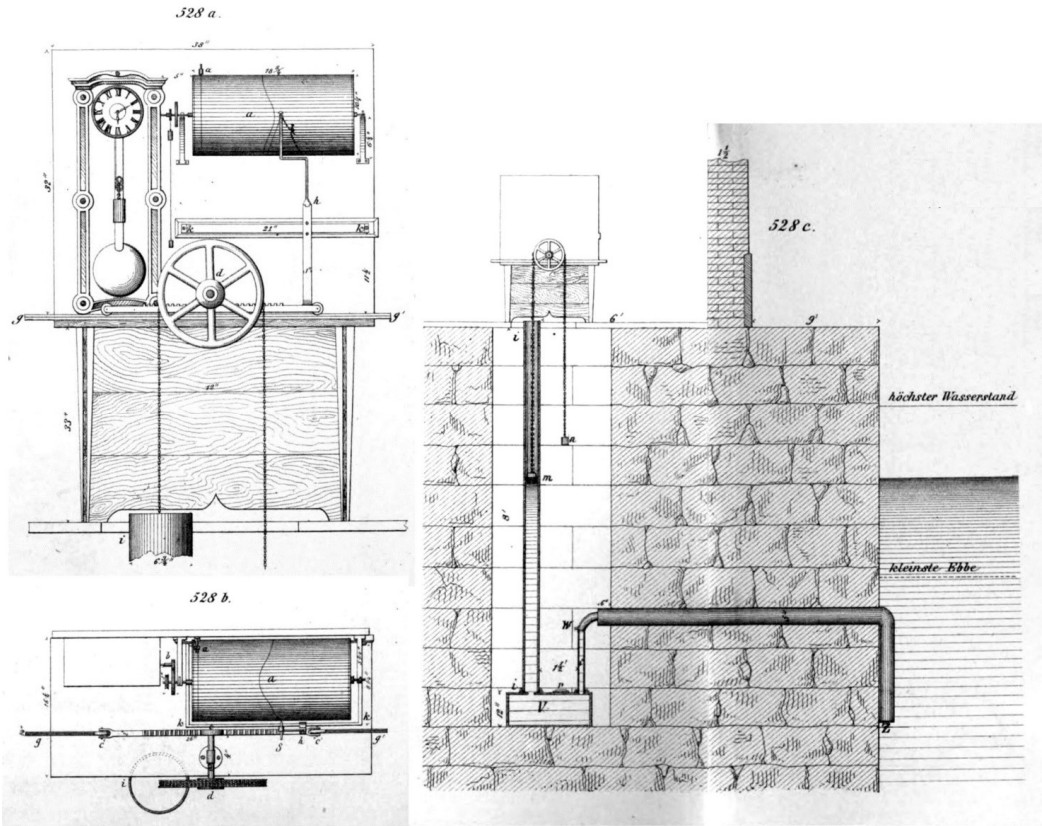

**Figure 2: The tide gauge installed in 1859. View of the instrument from the front (528a) and from above (528b); vertical section of the northwest end of the building hosting the tide gauge and the stilling well (528c). Mean high and low waters are indicated by 'höchster Wasserstand' and 'kleinste Ebbe', respectively. The figures are adapted from Chiolich-Löwensberg (1865; 1866).**

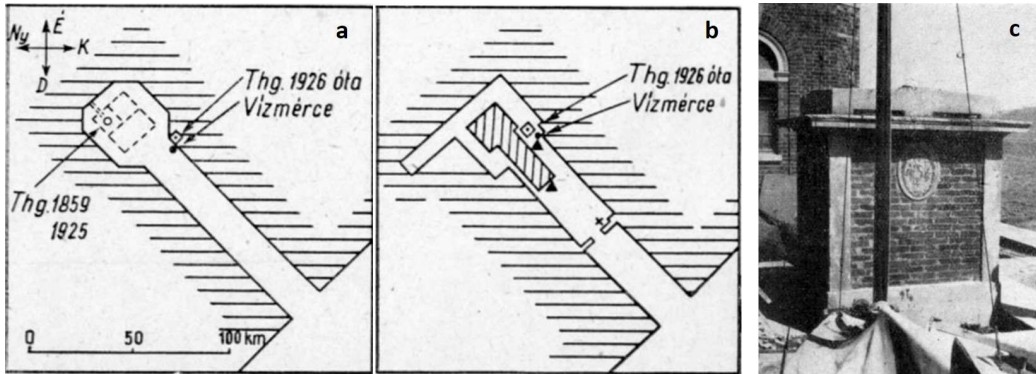

**Figure 3: (a) Map of Molo Sartorio before and (b) after the construction works of 1925–26. (c) Image of the new tide-gauge hut**
10 **probably in the early 1930's. '*Thg. 1859 1925*' ('Tide gauge 1859 1925') indicates the position of the old tide gauge (a) and '*Thg. 1926 óta*' ('Tide gauge from 1926') that of the new tide-gauge hut (a, b). '*Vízmérce*' indicates the hydrometer (a, b). North is upwards. Adapted from Bendefy (1958), who credited the photograph to S. Polli (Geophysical Institute of Trieste). (Note: the scale unit in (a) should be 'm'.)**



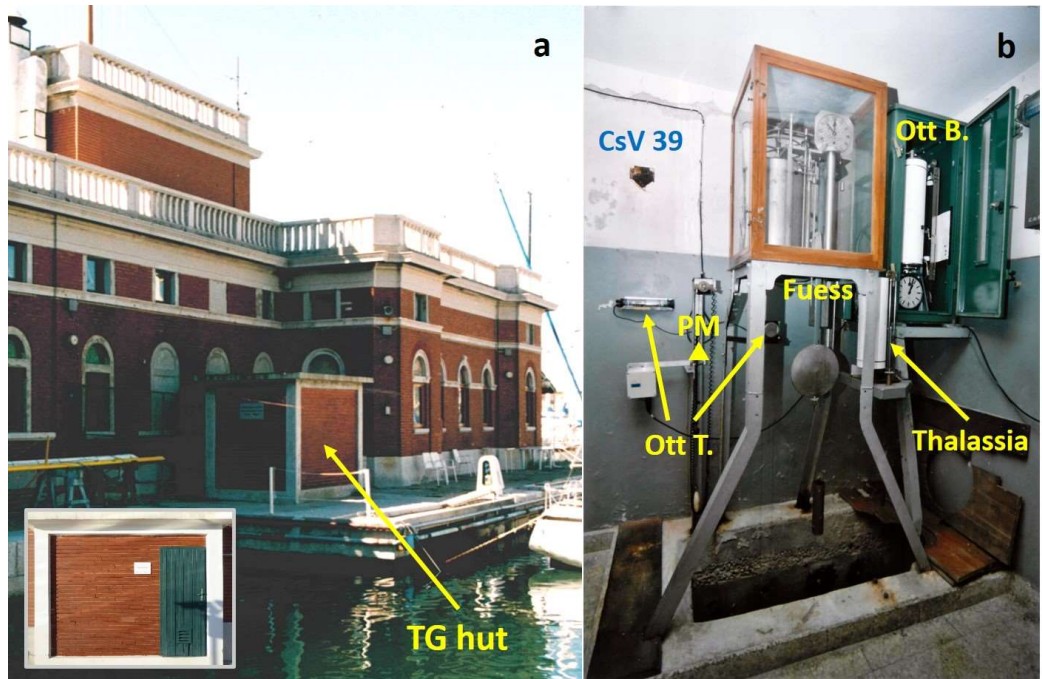

**Figure 4: (a) The tide-gauge hut in 2001; since 2004 only the wall indicated by the arrow is visible (see the inset) because the hut is enclosed in the main building, which was enlarged. (b) The inside of the tide-gauge hut in 2001. Four instruments are shown: Ott Thalimedes (1st) (Ott T.), Ott Büsum (Ott B.), Thalassia, and Fuess. PM is the TG CP, CsV 39 is a vertical benchmark no longer present. (Photographs by CNR-ISMAR, Trieste.)**

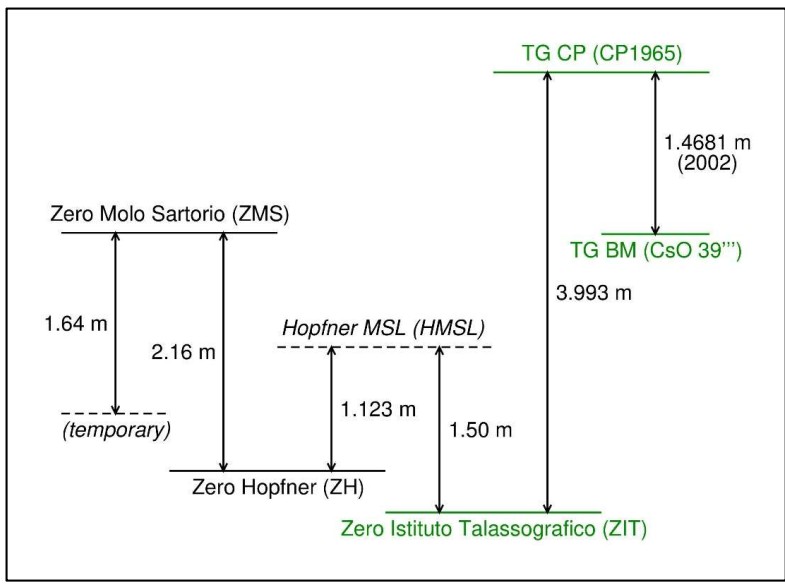

**Figure 5: The relationships between the tide-gauge Zeros and their connections to the tide-gauge contact point (TG CP) and benchmark (TG BM).**



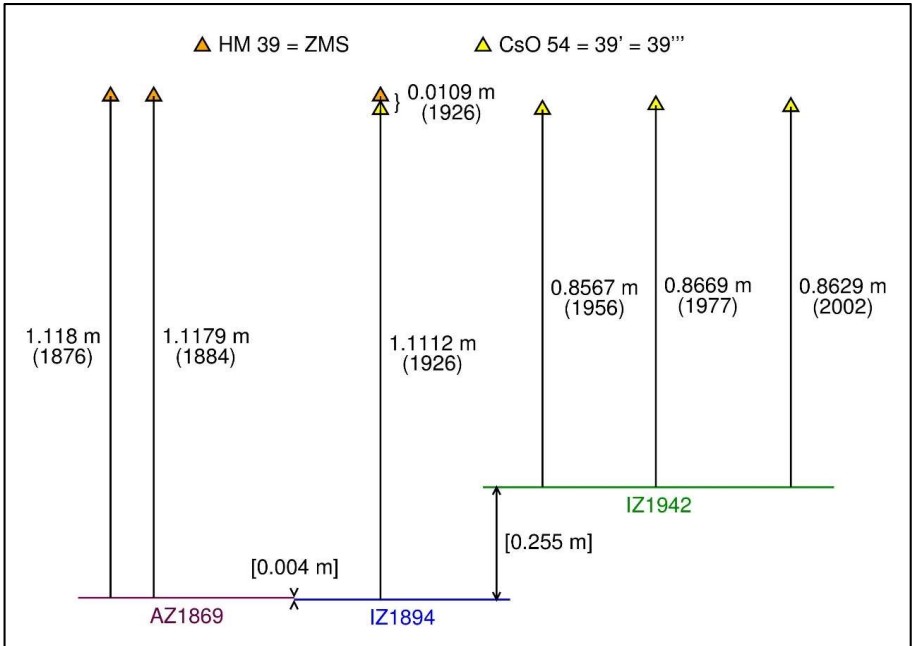

**Figure 6: Heights of the benchmarks used to build the composite time series; the year of the survey is in brackets. The Zeros of the levelling networks are shown: Austrian Zero of 1869 (AZ1869), Italian Zero of 1894 (IZ1894) and Italian Zero of 1942 (IZ1942). The differences between the Zeros have been estimated as discussed in Appendix A.**

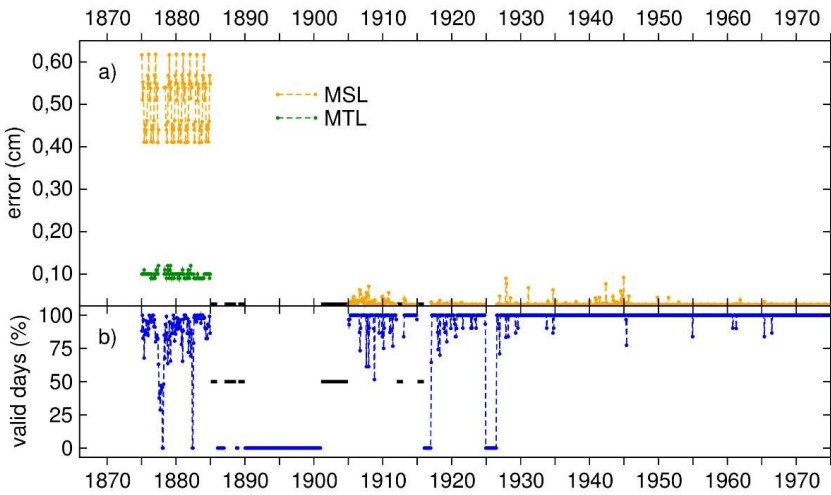

**Figure 7: (a) Monthly errors (centimetres) for 1875–1975; both the errors on MTL and MSL are shown for 1875–1889. (b) Monthly percentages of valid days for 1875–1975. After 1975 the error is always 0.03 cm and there are no missing days. The black segments indicate that the errors and percentages of valid days are unknown. The labels on the abscissa axis refer to the beginning of the year.**



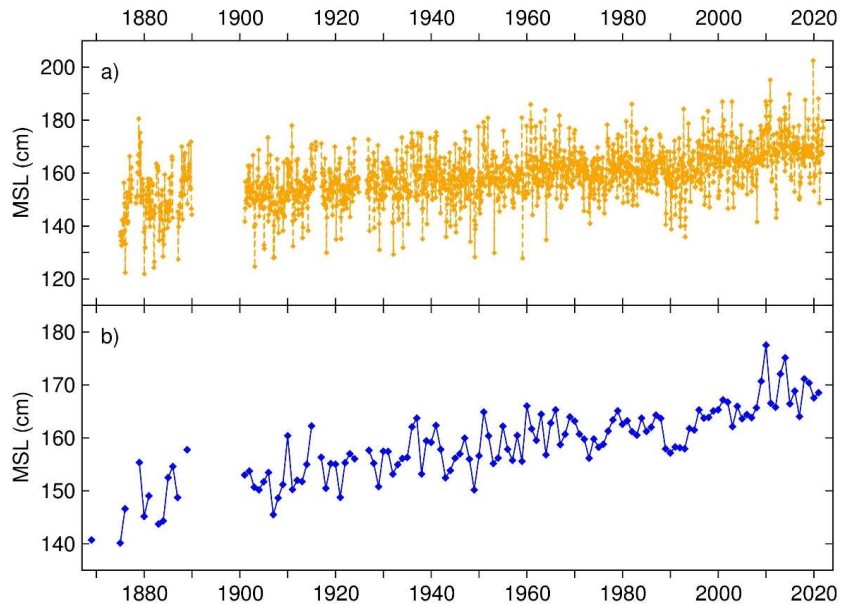

**Figure 8: (a) Monthly MSL and (b) Monthly MSL. The data are expressed in centimetres relative to the TG Zero (ZIT, see Fig. 5). The labels on the abscissa axis in (a) refer to the beginning of the year.**

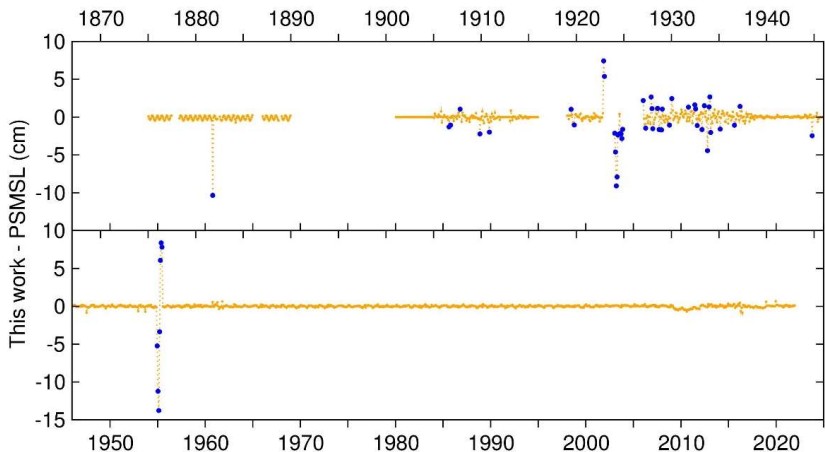

**Figure C1: Differences between the monthly MSLs obtained in this work and those in the PSMSL data bank (centimetres). The blue dots highlight differences greater than 1 cm, in absolute value. The labels on the abscissa axes refer to the beginning of the year.**

