# Peer review of "The sea level time series of Trieste, Molo Sartorio, Italy (1869–2021)"

_Earth System Science Data, 2022_

## Author Response (AR1)

Response to reviewers (**RC** = reviewer's comment, **AR** = author's response)

Please note that in the marked-up version, figures and tables can be found within the text.

Answers to Reviewer 1

**RC:** Page 1: please, clarify the sentence "Mazelle (1909-1917) reported the hourly data for 1905-1911." What do the years 1909-1917 mean?
**AR:** "Mazelle (1909-1917)" is the reference at page 14, line 8. It includes seven annual reports that contain the data from 1905 to 1911, that were published from 1909 to 1917. In order to avoid confusion:
1) The original text on page 1, lines 36-37 was changed to "The hourly data for 1905-1911 were reported in Mazelle (1909; 1910; 1911; 1912; 1913; 1914; 1917)." (see marked-up text, page 1, lines 36-38).
2) The collective reference (original text, page 14, line 8) was replaced by the individual items (see marked-up text, page 19, lines 39-43, and page 20, lines 1-3):
Mazelle, E.: Rapporto annuale dell'Osservatorio Marittimo 1905, Tip. Lloyd Austriaco, Trieste, Austria, 1909.
Mazelle, E.: Rapporto annuale dell'Osservatorio Marittimo 1906, Tip. Lloyd Austriaco, Trieste, Austria, 1910.
Mazelle, E.: Rapporto annuale dell'Osservatorio Marittimo 1907, Tip. Lloyd Austriaco, Trieste, Austria, 1911.
Mazelle, E.: Rapporto annuale dell'Osservatorio Marittimo 1908, Tip. Lloyd Austriaco, Trieste, Austria, 1912.
Mazelle, E.: Rapporto annuale dell'Osservatorio Marittimo 1909, Tip. Lloyd Austriaco, Trieste, Austria, 1913.
Mazelle, E.: Rapporto annuale dell'Osservatorio Marittimo 1910, Tip. Lloyd Austriaco, Trieste, Austria, 1914.
Mazelle, E.: Rapporto annuale dell'Osservatorio Marittimo 1911, Tip. Lloyd Austriaco, Trieste, Austria, 1917.

**RC:** Page 1, line 20: remove "and"
**AR:** Done (see marked-up text, page 1, line 20).

**RC:** Section 2.3: definition and linking among benchmarks are points of major relevance to build a consistent sea level record. The vertical distances between the different benchmarks are listed and represented. However, I miss some information on the horizontal distances among these points. Also, it would be informative to know how the levelling has been performed: which benchmarks are directly connected and how was the levelling carried out, and which benchmarks are intermediate connections. Also, there is no estimate of the levelling error. Is there information that would allow to compute this error, depending on the accuracy of the levelling for example? One way to do this is described in Marcos et al (2011), included in the reference list.
**AR:** In our case the consistency of the time series of relative sea level is guaranteed by the known relationships between the tide-gauge zeros, that can all be referred to Zero Molo Sartorio. I am afraid that I can provide only part of the information requested by the reviewer. The technical details about the levelling surveys of the 19$^{th}$ century could not be found. With regard to the surveys since 1926, information is not fully public because of restrictions imposed by the Italian Military Geographic Institute, which carried out the surveys. As regard to the distances between benchmarks, new pieces of text were inserted as follows.

Original text, page 3, line 22 (sect. 2.3.1): "… The horizontal distance between HM 1 and HM 39 was 52 m along the levelling line (MGI, 1885; 1892). …" (see marked-up text, page 6, lines 17-18).
Original text, page 3, line 33 (sect. 2.3.2): "CsO 53 was connected with CsO 52, located 879 m away. Subsequently, the survey involved CsO/CsV 53A, 42 m away from CsO 53, and CsO 54, 26 m away from CsO 53A (IGMI, 1926)" (see marked-up text, page 7, lines 11-13).
Original text, page 4, lines 2-3 (sect. 2.3.2): "Both CsO 39 and CP1965 are 6 m away from CsO 39' (formerly 54). CsV 39 is just above CP1965." (see marked-up text, page 7, line 22).

**RC:** Page 5: "The complete time series of hourly data from 1905 to 2021 obtained in this work is available in Raicich (2022)". The data set in the reference indicates that the period is 1939-2018
**AR:** The link in Raicich (2022) (page 15, line 1) leads to the web page https://www.seanoe.org/data/00516/62758/, which gives access to two datasets. The first one is for 1869-2021, which includes the hourly data for 1905-2021 and is freely accessible, the second one is for 1939-2018, which is now outdated and only accessible on request. I agree that this might be confusing. Unfortunately, SEANOE's policy is to keep the older versions. In order to clarify this point:
The original text on page 5, line 4, was rephrased as "The complete time series of hourly data from 1905 to 2021 obtained in this work is included in the data set for 1869-2021 available in Raicich (2022)." (see marked-up text, page 9, line 4).

Answers to Reviewer 2

**RC:** However, the author might consider adding more information to further increase the interest of this study. My suggestions substantially concern the time behaviour of the trend and whether a recent acceleration can be identified.
The author writes that (Line 11)  "a significant acceleration of 0.008±0.004 mm yr-2 was estimated from the inverse-barometer-corrected sea level time series". This sentence suggests that sea level rate accelerated uniformly in time. Is this the best way of summarizing the changes of sea level rise rate or there is evidence of a change of the trend in the last 3 or 4 decades?
I suggest adding a discussion whether the acceleration of sea level in the last decades (which is supported by satellite altimetry at global scale) is supported also by this regional time series. If we consider 50 years long period (e.g. 1972-2021, 1922-1971, 1872-1921) or 30 years long periods (e.g. 1872-2001, …, 1992-2021) do we see trends gradually increasing or a sharp difference between the last period and the previous ones?
**AR:** The original sentence on page 1, lines 11-12 (in the abstract), does not suggest that the acceleration is uniform, it just summarizes a property of the whole time series. Details are provided in Section 4, where MSL rise rates are also reported, both for the whole time series, namely 1.5 mm/yr, and in the altimetry period, that is 3.0 mm/yr. This answers the question whether a rate change, i.e. acceleration, has been observed recently.
Because this is a 'data paper', I would not like to extend the discussion to scientific issues as the detection of the transition epoch. The main reason is that the transition is more or less smooth depending on the length of the time period used to estimate the trends. Moreover, in the present case, a complication comes from the peculiar behaviour of the Mediterranean sea level between the mid 1960's and the early 1990's (see page 6, lines 28-30). Let the trends be estimated in time windows of N years. If we assume that the sea level steadiness started (for instance) in 1965 and ended in 1993, all the trend estimates from [1965-N+1,1965] to [1993,1993+N-1] are affected by the steady sea level. If N = 30, even the latest period is not

completely unaffected. Clearly, this is what the observations at Trieste tell us. The point is that a comparison with the global ocean is not straightforward and I think that such an issue deserves a dedicated study which goes beyond the scope of the paper.

However, to partly address the question, the acceleration was also estimated removing the 'critical' period.

In the original text, page 7, after line 22 (end of sect. 4), the following text was added:

"To assess how the steady sea level period affects the trend and acceleration estimates, we analysed an IB-corrected time series from which the annual means from 1967 to 1995 were removed. As a result, the linear trends are 1.55±0.14 mm yr$^{-1}$ for 1869-2021 and 1.56±0.16 mm yr$^{-1}$ for 1905-2021, while the accelerations are 0.004±0.005 mm yr$^{-2}$ for 1869-2021 and 0.004±0.006 mm yr$^{-2}$ for 1905-2021, respectively. In both cases accelerations are statistically not significant at p = 0.05." (see marked-up text, page 12, lines 19-22).

**RC:** Minor comment: Figure 8: (a) Monthly MSL and (b) Monthly MSL… it seems something is wrong here, likely b) shows annual mean msl (consistently with the text)

**AR:** Yes, the reviewer is right. The mistake was corrected: "(b) annual" instead of "(b) Monthly" (see marked-up text, page 11, line 1 of the figure caption).

Additional corrections

A few typos were discovered and corrected, and a reference was updated.

Original text, page 2, line 23: '30' instead of '20' (see marked-up text, page 4, line 18).

Original text, page 3, line 31 and page 17, Table 2, 8[th] line: 'CsV 53A' instead of 'CsV 53' (see marked-up text, page 7, line 9, and page 6, Table 2, 8[th] line, respectively).

Original text, page 4, lines 5-6: '(IGMI, 2009)' instead of '(Zambon, … 2008)' (see marked-up text, page 7, lines 25-26). The relevant reference was inserted in the original text, page 13, after line 35(see marked-up text, page 19, lines 22-23).

Original text, page 9, line 14 (Eq. A16): 'AZ1869' instead of 'IZ1869' (see marked-up text, page 14, line 22-23).

Original text, page 14, line 15: '58-120' instead of '59-120' (see marked-up text, page 20, line 10).